# UAV-Based Low Altitude Remote Sensing for Concrete Bridge Multi-Category Damage Automatic Detection System

Han Liang ⬤, Seong-Cheol Lee *⬤ and Suyoung Seo ⬤

Department of Civil Engineering, Kyungpook National University, Daegu 41566, Republic of Korea; shto520@knu.ac.kr (H.L.); syseo@knu.ac.kr (S.S.)
* Correspondence: seonglee@knu.ac.kr; Tel.: +82-53-950-5608

**Abstract:** Detecting damage in bridges can be an arduous task, fraught with challenges stemming from the limitations of the inspection environment and the considerable time and resources required for manual acquisition. Moreover, prevalent damage detection methods rely heavily on pixel-level segmentation, rendering it infeasible to classify and locate different damage types accurately. To address these issues, the present study proposes a novel fully automated concrete bridge damage detection system that harnesses the power of unmanned aerial vehicle (UAV) remote sensing technology. The proposed system employs a Swin Transformer-based backbone network, coupled with a multi-scale attention pyramid network featuring a lightweight residual global attention network (LRGA-Net), culminating in unprecedented breakthroughs in terms of speed and accuracy. Comparative analyses reveal that the proposed system outperforms commonly used target detection models, including the YOLOv5-L and YOLOX-L models. The proposed system's robustness in visual inspection results in the real world reinforces its efficacy, ushering in a new paradigm for bridge inspection and maintenance. The study findings underscore the potential of UAV-based inspection as a means of bolstering the efficiency and accuracy of bridge damage detection, highlighting its pivotal role in ensuring the safety and longevity of vital infrastructure.

**Keywords:** object detection; remote sensing; attention mechanism; bridge damage; UAV inspection system





## 1. Introduction

Concrete bridge damage detection has become a topic of significant interest in recent years due to the aging infrastructure of many bridges globally [1]. Various factors such as natural disasters, aging, corrosion, fatigue, and overloading can cause bridge damage [2,3]. Traditional inspection methods are time-consuming, costly, and potentially dangerous. Non-destructive testing methods such as vibration-based [4,5], acoustic emission [6,7], and imaging-based techniques have been explored to detect, locate, and quantify damage in bridge structures. However, manual imaging-based methods have limitations such as limited accessibility, low accuracy, and long inspection time. Using unmanned aerial vehicles (UAVs) offers several benefits, including the ability to access difficult or dangerous areas, capture images from different angles and heights, and reduce inspection time and cost while enhancing safety [8,9]. Designing a complete bridge damage inspection system using UAVs offers a promising solution to improve inspection accuracy, efficiency, and safety while ensuring bridge longevity.

For concrete bridge damage detection, traditional image processing algorithms are widely used. These algorithms typically involve a series of steps to process acquired images and identify potential damage areas. Feature-based algorithms have shown high accuracy in identifying specific types of damage, such as cracks in concrete structures, composite concrete bridge damage, and various types of reinforced concrete damage, using multiple image features and support vector machine (SVM) classifiers [10–13]. To enhance bridge damage detection, various image feature processing methods have been combined,

such as Hough transform, Laplacian of Gaussian (LoG) weighted, dilation, grayscale, Canny edge detection, and Haar wavelet transform. Moreover, automated surface crack detection robots have been developed, utilizing a sectoring method to capture the concrete surface systematically, and the Haar-trained cascade object detector is used for surface crack classification, effectively identifying surface crack defects [14]. However, the effectiveness of these algorithms depends on selecting appropriate features, and they may not capture more complex or subtle forms of damage.

While traditional image processing algorithms can be useful in screening potential damage areas, their limitations emphasize the necessity for more advanced and sophisticated methods. Relying heavily on the accuracy of selected features, templates, or models can be challenging to develop and maintain, particularly for large or complex concrete bridge structures. Consequently, there is a growing demand for more advanced and sophisticated methods, such as deep learning techniques, to enhance the accuracy, reliability, and efficiency of bridge inspections.

Deep learning techniques have shown promise in enhancing the accuracy, reliability, and efficiency of concrete bridge inspections by identifying damage with high precision. One commonly used approach is semantic segmentation, which can differentiate damaged or deteriorated areas such as cracks, spalls, or corrosion from surrounding healthy material. Several studies have proposed deep learning-based models for detecting bridge damage using semantic segmentation. For example, Wang et al. [15] proposed an improved end-to-end bridge crack detection model based on the Inception-Resnet-v2 algorithm, achieving high performance with an accuracy of 99.24%, recall of 99.03%, F-measure of 98.79%, and FPS of 196. Similarly, Li et al. [16] proposed a flexible crack identification system using sliding window technology to construct a dataset and a trainable context encoder network with recurrent convolutional neural network (R-CNN) and region max-pooling (RMP) to enhance the accuracy and stability of crack segmentation, achieving an accuracy of 98.62% and mIoU of 80.93%. Additionally, Li et al. [17] proposed the skip-squeeze-and-excitation networks (SSENets) model, including the skip-squeeze-excitation (SSE) and atrous spatial pyramid pooling (ASPP) modules, achieving a 97.77% accuracy in detecting bridge cracks. Furthermore, Jiang et al. [18] proposed HDCB-Net, a deep learning-based network with a hybrid dilated convolutional block for pixel-level crack detection on concrete bridges. The proposed method is accurate for detecting blurred cracks and efficient for fast crack detection, and a public dataset has been established for further research. Xu et al. [19] proposed a lightweight, accurate, and robust semantic segmentation method for complex structural damage recognition in actual concrete bridges, achieving significant improvements over existing lightweight crack segmentation models. Finally, Li et al. [20] presented the design of a crack detection network named DBR-Net, which combines STDC-Net and a refinement network to balance speed and accuracy, enabling real-time crack detection. However, these representative semantic segmentation methods cannot simultaneously localize and classify multiple types of bridge damage. Moreover, while these methods have good processing speed and accuracy, their application in practical bridge damage detection systems is limited due to the requirement for vertically-taken images close to the damage.

Numerous studies have proposed solutions for detecting and classifying multiple types of damage to concrete bridges. Zhang et al. [21] proposed an improved algorithm using YOLOv3, MobileNets, and CBAM to detect surface damage on bridges. Another study [22] proposed an improved YOLOv4 network with a pruning technique and Evo-Norm-S0 structure to identify various types of concrete damage, including plant, expansion joint, watermark, surface joint, manual painting, and structural cracks. This model outperformed other leading algorithms, such as SSD300, YOLOv3, and YOLO X-L, in detecting concrete cracks with high accuracy and fast calculation speed. Teng et al. [23] proposed an improved YOLOv3 network for bridge surface defect detection, which was compared with YOLOv2 and Faster RCNN and further improved using transfer learning and data augmentation. Finally, Wan et al. [24] introduced a novel deep learning model called

BR-DETR, based on detection transformers (DETR), for detecting damage on the surface of bridge structures, including spalls, rebars, and cracks. This model outperformed the DETR model in detection performance with increased mean average precision (mAP) and recall by using the copy-paste data augmentation method, deformable conv2d, convolutional project attention, and locally-enhanced feed-forward.

However, manual image acquisition for these methods has limitations. It is time-consuming, labor-intensive, and may fail to capture hard-to-reach or concealed areas where damage is invisible to the naked eye. It can also pose safety risks to inspectors, especially in high or inaccessible areas, resulting in additional costs and complexities. Moreover, the frequency of manual photography may not be sufficient to detect damage promptly, which could pose potential safety hazards if not identified and repaired promptly.

Given the various challenges in related work for bridge damage detection, and the need for practical automated inspections, this paper proposes a system for detecting multi-category bridge damage using low-altitude remote sensing by UAV. The process operation of the proposed system is shown in Figure 1, and the main contributions of this work are as follows:

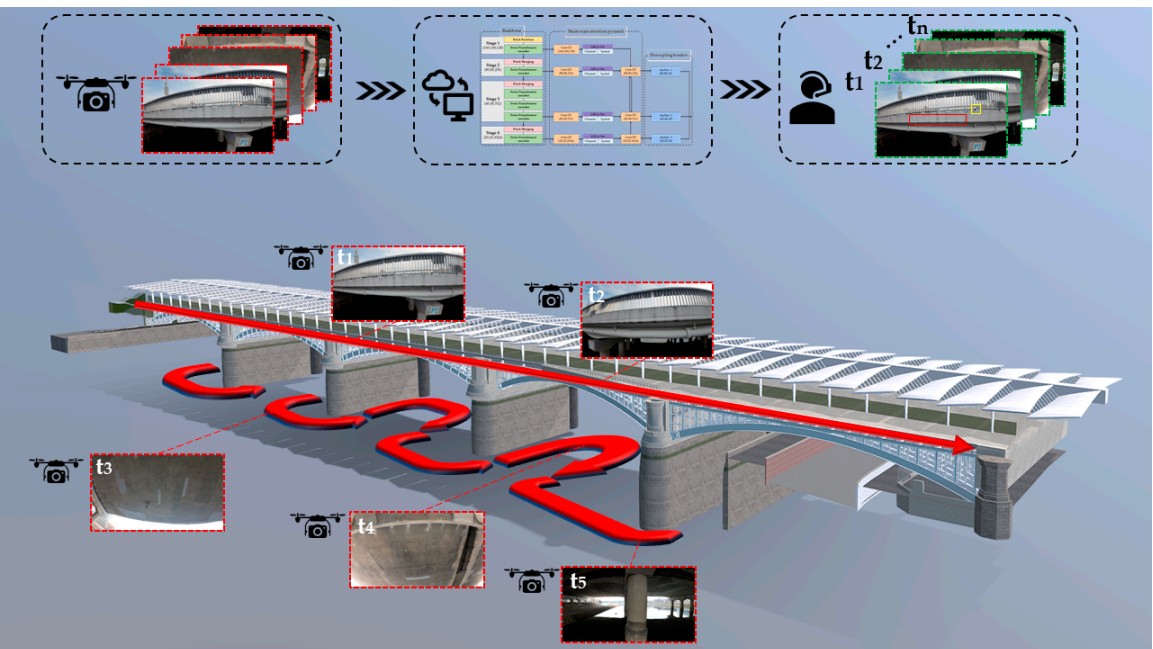

**Figure 1.** Flowchart of the proposed UAV-based low-altitude remote sensing system for detecting multiple types of damage to bridges.

Firstly, the proposed UAV-based inspection system addresses the limitations of manual image acquisition, including time-consuming, labor-intensive, and high-risk inspections. The advantages of UAVs, such as high flexibility, mobility, and accessibility, enable efficient and comprehensive inspections of bridges, including hard-to-reach areas, while minimizing risk to inspectors.

Secondly, an end-to-end fully automated bridge multi-type damage detection network is proposed. This approach incorporates an efficient Swin Transformer as the backbone network, and a multi-scale pyramidal attention network for feature fusion, which adapts to the target size changes triggered by the change in viewpoint of the UAV. Additionally, the improved attention mechanism for long-range dependencies addresses the limitations of previous models, enhancing the performance of the network.

Thirdly, experimental results demonstrate the effectiveness of the proposed method, outperforming other classical models for the detection of five types of bridge damage, including corrosion stains, cracks, efflorescence, exposed bars, and spallation. With a mAP of 61.27% on the open-source dataset, the proposed method offers a significant improvement in accuracy and efficiency over other widely used classical models. This paper contributes

to the field of bridge maintenance by integrating cutting-edge technologies such as drones and deep learning to improve the efficiency and accuracy of the inspection process. The proposed method offers a comprehensive, efficient, and automated approach to bridge damage detection, providing significant benefits to the maintenance and management of bridge infrastructure.

Section 2 presents the methods and materials used in this study, including the design of the network for UAV inspection systems, the field experiments for bridge damage detection, and the evaluation indices and training strategies. Section 3 presents the results, including the comparison of the performance of each backbone network, the comparison of the attention modules, and the comparison of performance with other classical networks. In Section 4, the discussion interprets and contextualizes the results, providing insights into the limitations and potential implications of the proposed approach. Finally, Section 5 concludes the article by summarizing the main contributions and discussing future research directions.

## 2. Materials and Methods

### 2.1. Network for UAV Inspection Systems Designed for Bridge Damage Detection

The proposed design for detecting bridge damage involves a network that comprises three primary steps: a backbone feature extraction network based on the Swin Transformer encoder, a multi-scale attention pyramid network, and anchor-based decoupling headers, as illustrated in Figure 2.

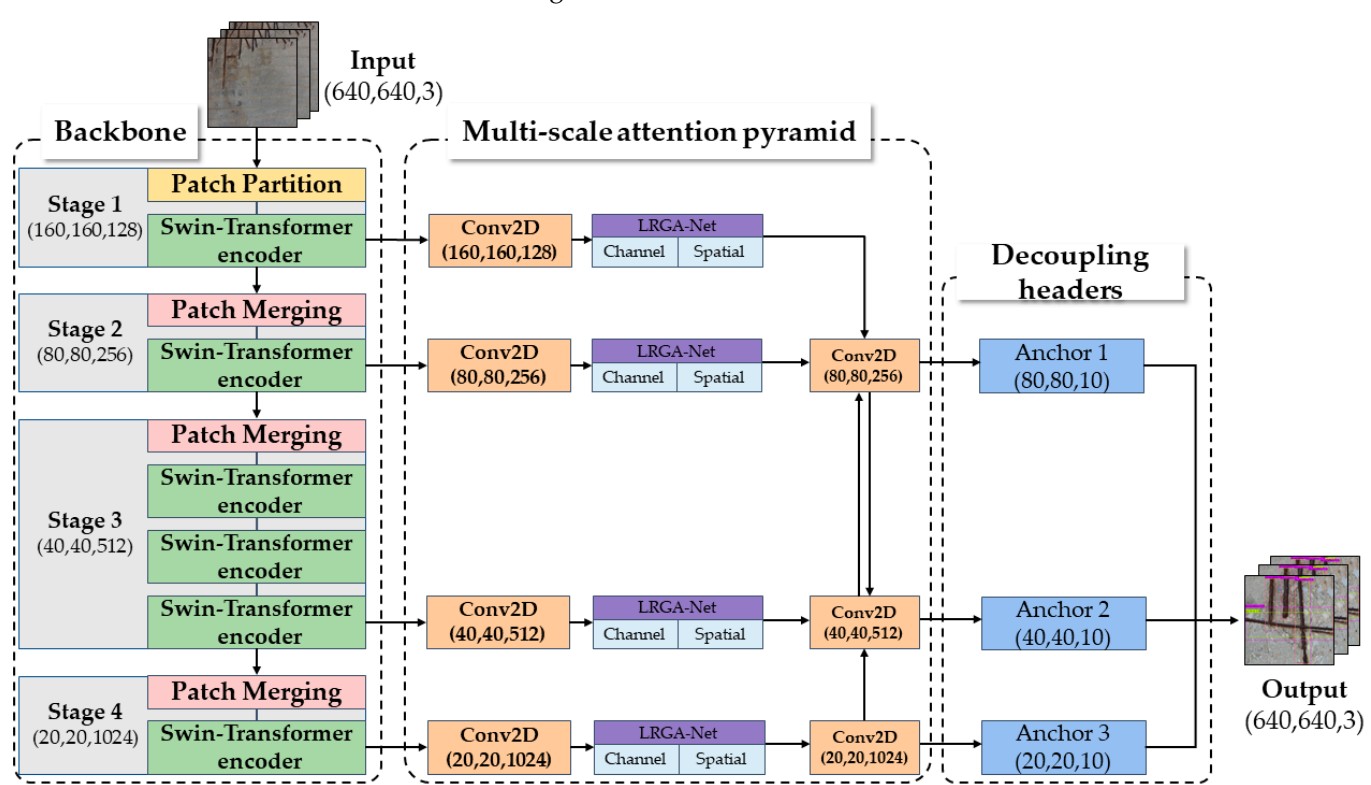

**Figure 2.** The bridge damage detection network is applied to UAV inspection systems.

2.1.1. Backbone Network Designed Based on Swin Transformer Encoder

The Swin Transformer encoder consists of several key components, including the windows multi-head self-attention (W-MSA) module [25], the shifted windows multi-head self-attention (SW-MSA) module [26], layer normalization (LN), and the multi-layer perceptron (MLP).

The W-MSA module is a standard component of the transformer and is responsible for capturing the relationships between tokens in the input sequence, as shown in Figure 3a. The novel SW-MSA module, on the other hand, aims to improve the representation of the input sequence by considering dependencies between tokens at different scales. The

SW-MSA module divides the input sequence into multiple windows of varying sizes and computes the attention scores between the tokens in each window rather than between every pair of tokens in the input sequence, as is done in the W-MSA module, as shown in Figure 3b.

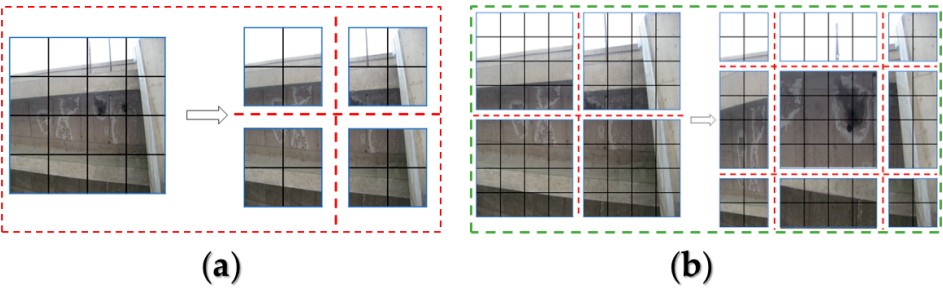

**Figure 3.** The illustration of (**a**) W-MSA operation and (**b**) SW-MSA operation.

The Swin Transformer encoder has layers, each of which includes the W-MSA, SW-MSA, LN, and MLP modules, as shown in Figure 4. LN normalizes the inputs to each module while the MLP implements the feed-forward network in the transformer. The input sequence is first processed by the W-MSA module to capture the dependencies between tokens, then by the SW-MSA module to improve the representation of the input at different scales, and finally by the MLP module to generate the final representation of the input sequence.

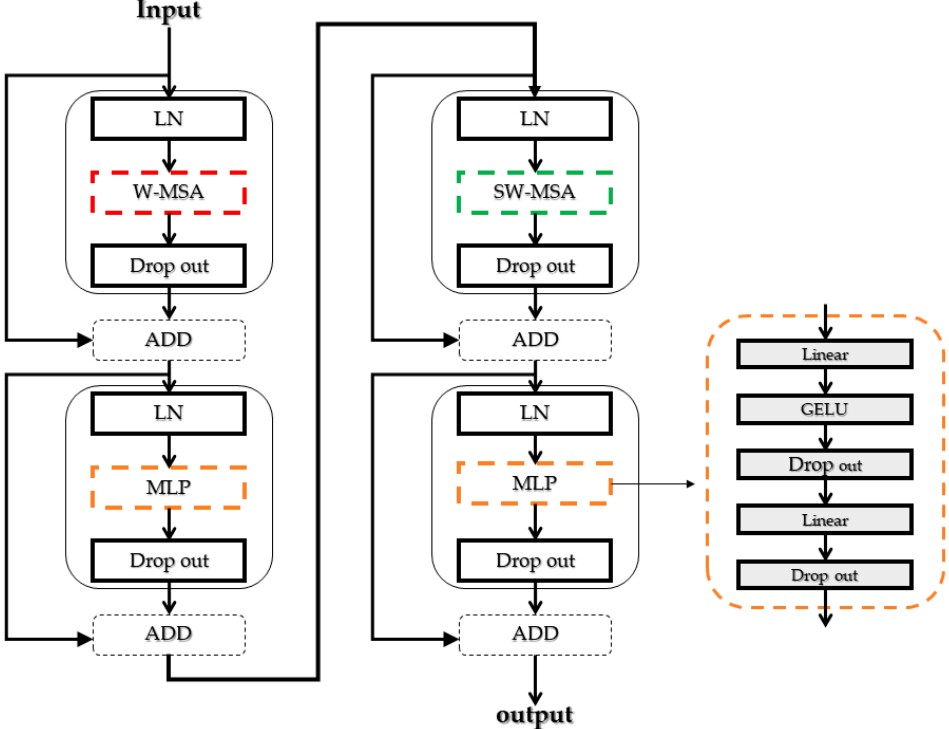

**Figure 4.** The Swin Transformer encoder forms the backbone network and comprises the W-MSA, SW-MSA, LN, and MLP connected in series.

Compared to a standard transformer, the Swin Transformer introduces several optimizations. One of these optimizations is local attention, which applies the attention mechanism only to a small portion of the input sequence instead of the entire sequence. This results in improved accuracy, especially for sequences with long-range dependencies. Additionally, the SW-MSA module reduces computational complexity by considering the dependencies between tokens at different scales rather than between every pair of tokens

in the input sequence. Furthermore, the Swin Transformer uses a more efficient LN and feed-forward network, leading to a reduction in computational resources compared to the standard transformer. As a result, using a feature layer extracted by the Swin Transformer in a target detection model could result in a more efficient and scalable model capable of handling larger input sequences with fewer computational resources.

The backbone uses the patch embedding (PE) and patch merging (PM) modules of its encoder. Patch embedding is the process of projecting each patch of the input image into a feature space using a linear transformation.

The input image is first partitioned into patches, where each patch consists of 4 × 4 adjacent pixels, as shown in Figure 5. The patches are then flattened, and each pixel is represented by its R, G, and B values, resulting in a flattened image of shape (160, 160, 48) from the original image of shape (640, 640, 3). The linear embedding layer then adjusts the number of channels of each pixel to 128, and the Swin Transformer encoder extracts the features of the first stage with a shape of (160, 160, 128).

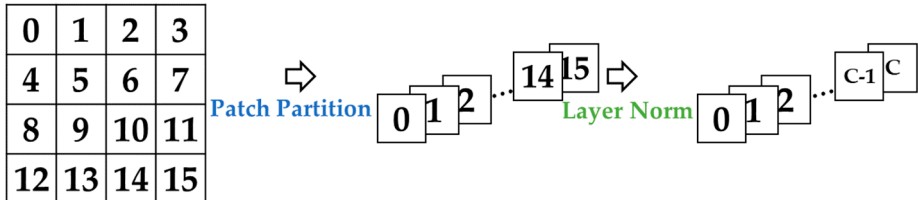

**Figure 5.** Sample of PE module processing input images.

Next, the PM module takes the remaining three stages and changes the number of channels in the feature layers. PM aggregates pairs of adjacent patches, and the resulting feature vector represents the merged patches at a finer scale. The module divides a 4 × 4 feature map into 2 × 2 patches, stitches the pixels in the same position (same color) in each patch to form four feature maps, and then concatenates the four feature maps together in the channel direction. The concatenated feature maps are then passed through an LN layer, and the number of channels of the feature map is halved by a linear transformation, as shown in Figure 6. This process results in feature layers with three sizes, namely, stage 1, stage 2, stage 3, and stage 4, with shapes of (160, 160, 128), (80, 80, 256), (40, 40, 512), and (20, 20, 1024), respectively.

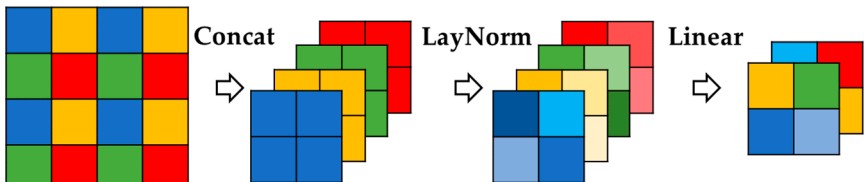

**Figure 6.** Sample of PM module providing downsampling details for a channel.

By utilizing different scales of feature layers, the fused feature network can effectively capture object details across a wide range of scales, resulting in improved accuracy and robustness of the model in detecting targets at different distances. Additionally, fusing multi-scale networks can help address the issue of scale imbalance in target detection, ensuring that objects at all scales are represented equally in the training data.

### 2.1.2. Multi-Scale Attention Pyramid Network

Multi-scale attention pyramid networks have the potential to significantly improve the accuracy and efficiency of target detection, especially in scenarios where target size varies, and there may be significant background clutter or noise during UAV detection. The network shown in Figure 2 involves combining feature layers from different stages of the backbone network and weighting them using an attention mechanism, which helps to filter out irrelevant information and highlight important features. By doing so, the network is able to efficiently capture object details across a wide range of scales and handle occlusions

and background clutter, which ultimately improves its stability for different scenes and challenges. The use of multi-scale feature layers in the network allows for the detection of targets of varying sizes, which is essential for many applications. Additionally, the attention mechanism used in the network can help to improve the efficiency and effectiveness of the network, making it more useful for target detection, where background clutter or noise can be problematic.

In previous research, we replaced the fully connected layer in the channel attention module of CBAM [27] with 1D convolution [28]. By incorporating the residual structure, we developed an enhanced network called the lightweight residual convolutional attention network (LRCA-Net), which effectively captures cross-channel interaction information while reducing the total number of module parameters to improve overall efficiency [29]. However, upon evaluating the spatial attention module of the LRCA-Net, with the structure shown in Figure 7, we identified some limitations, which led us to propose an improved approach. The current spatial attention module utilizes a method of concatenating and aggregating two pooling results to form a feature of shape $H \times W \times 2$, which is later convolved with a standard $7 \times 7$ convolution layer.

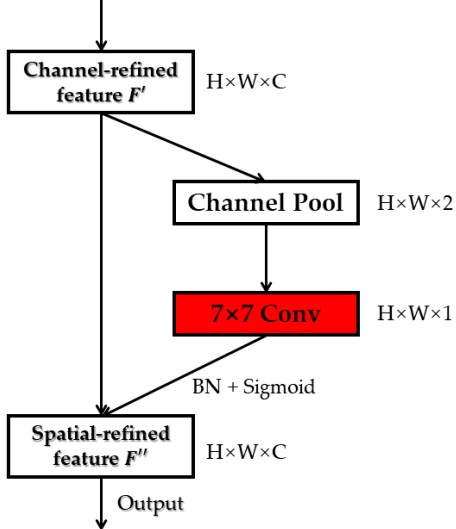

**Figure 7.** The LRCA-Net's spatial attention module restricted by the convolutional kernel size, which limits the perceptual field to local feature information only.

The major drawback of this approach is that it is limited by the size of the convolution kernel, which only contains local feature information and cannot synthesize global information shown in Figure 8.

As an improvement over the spatial attention module of LRCA-Net, this paper proposes the lightweight residual global attention network (LRGA-Net). The main idea of LRGA-Net is shown in Figure 9. The standard spatial averaging pooling operation is performed on the input feature $F'$, where the input 2D tensor is x with a size of $H \times W$ and a number of input channels of $C$. The pooling space is $h \times w$, the output 2D tensor is $y$, and the size is shown in Equation (1).

$$y = H_0 \times W_0 \quad (H_0 = \frac{H}{h}, W_0 = \frac{W}{w}),  \tag{1}$$

The average pooling process is represented by Equation (2),

$$y_{i_0,j_0} = \frac{1}{h \times w}\sum_{0 \leq i \leq h}\sum_{0 \leq j \leq w} x_{i_0 \times h+i,j_0 \times w+j},  \tag{2}$$

where $0 < i_0 < H_0$, $0 < j_0 < W_0$, and each position of $y$ corresponds to a window of $h \times w$.

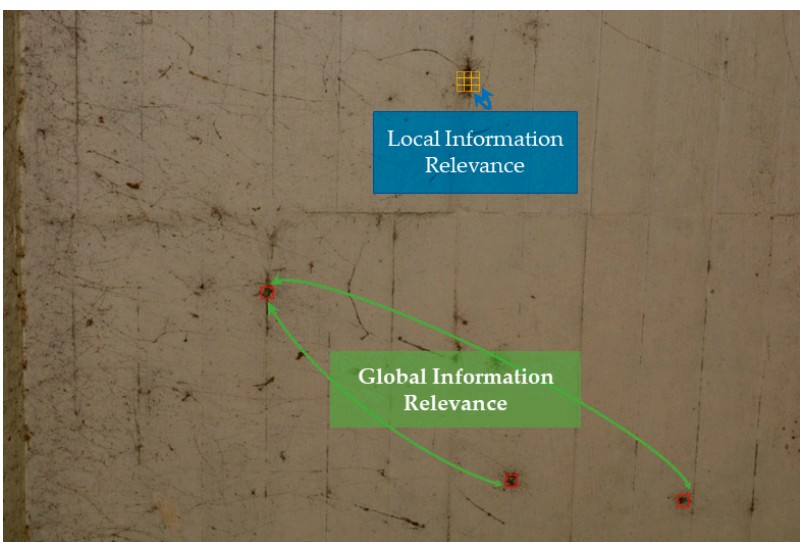

**Figure 8.** Local dependencies are captured through convolutions (indicated in yellow), while long-distance connections (indicated in red) capture global dependencies.

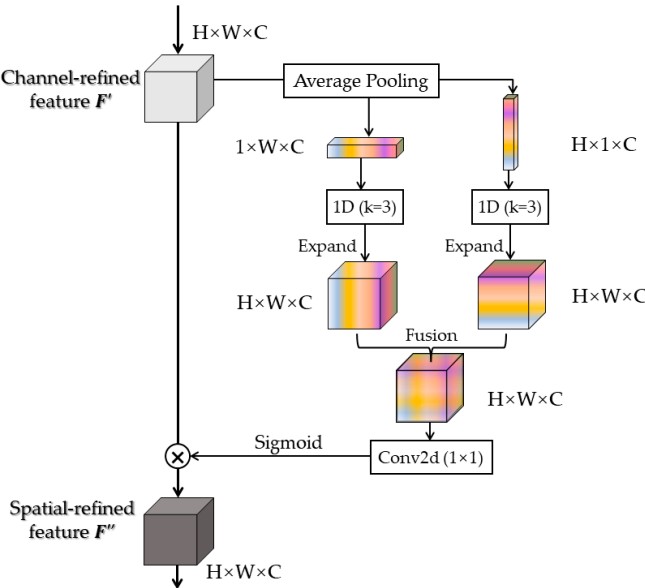

**Figure 9.** The modified spatial attention module.

This operation is performed separately for the horizontal and vertical directions to obtain two long and narrow feature layers. The resulting feature layers are then convolved with a 1D convolution of kernel size 3 for modulation of the current position and its collinear features. The two resulting feature layers are then fused using Equation (3).

$$y_{c,i,j} = 1D\left(y_{c,i}^h\right) + 1D\left(y_{c,j}^w\right) \tag{3}$$

where $y \in R^{C \times H \times W}$, 1D represents a 1D convolution with a convolution kernel of size 3. The final spatial-refined feature $F''$ is obtained using Equation (4).

$$F'' = Scale(F', \sigma(Conv2d(y))) \tag{4}$$

where $Scale(,)$ represents the element multiplication, $\sigma$ represents the sigmoid function, and $Conv2d$ represents the convolution with a kernel of $1 \times 1$.

The improved spatial module added to the overall attention mechanism is depicted in Figure 10. It enhances the dependencies among long-range features while remaining

lightweight due to the use of 1D convolution. Furthermore, the shape of the input and output remains the same, making it easy to insert LRGA-Net into the network.

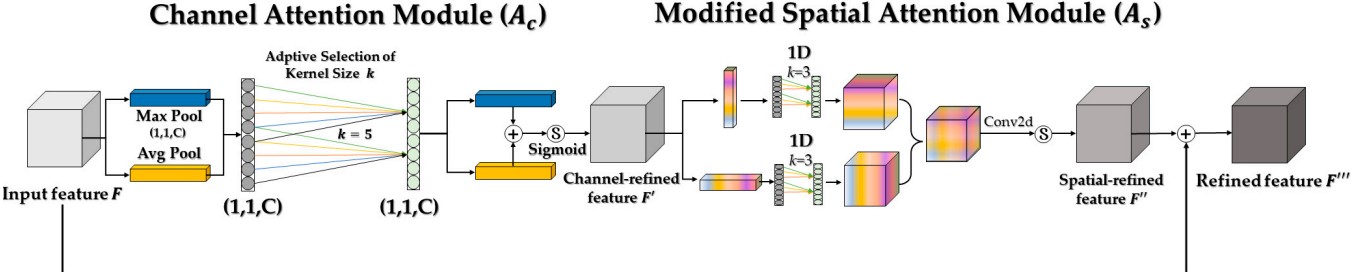

**Figure 10.** The overall architecture of LRGA-Net.

2.1.3. Anchor-Based Decoupling Headers

The end output of the network uses an anchor-based approach to determine the bounding box of the target and its confidence level. For each of the three output feature layers, nine prior boxes are predefined with different sizes to be adjusted during prediction. The mask of the anchor is generated based on K-means clustering [30] analysis of the object's bounding boxes in the training dataset.

The calculation of the anchor boxes' dimensions based on the outcomes of the K-means clustering analysis is visually depicted in Figure 11. In Table 1, the masks for the anchor boxes are presented. Subsequently, these proposals are assessed by considering both the predicted class probabilities and their associated bounding box regression outcomes. To ensure the selection of the most suitable candidate boxes while eliminating any overlapping or low-confidence boxes, the non-maximum suppression algorithm is applied.

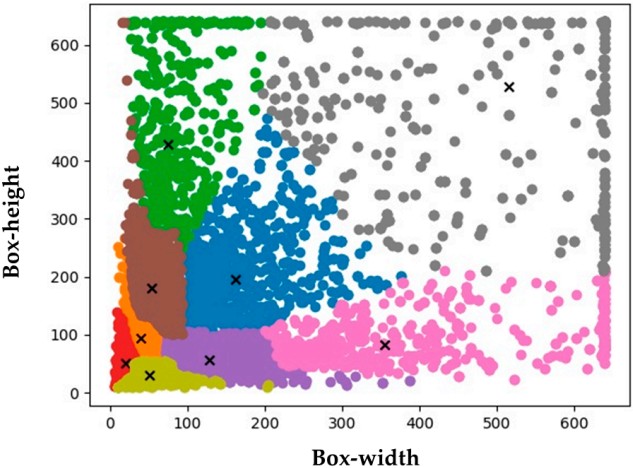

**Figure 11.** Normalized height and width clusters of bounding boxes obtained through K-means clustering.

**Table 1.** The parameters for setting the anchor mask obtained through the K-means clustering process.

| Anchor Layer | Anchor 1 | Anchor 2 | Anchor 3 |
|---|---|---|---|
| Anchor Size (Width, Height) | (20, 50)<br>(50, 30)<br>(39, 94) | (129, 56)<br>(53, 179)<br>(355, 83) | (161, 195)<br>(75, 428)<br>(514,527) |

*2.2. UAV Detection of Bridge Damage Field Experiments*

The experimental site selected for remote sensing inspection was the Kyungdae Bridge, which is located in the North District of Daegu, South Korea (35° N 128° E), as illustrated in Figure 12a. The bridge is a reinforced concrete structure with a total length of 130 m, built in 1993, and has been open for 30 years. For the inspection, a DJI Avata UAV was utilized,

as shown in Figure 12b, and the path taken during the inspection is roughly depicted in Figure 12c. The inspection focused on the concrete surface at the bottom of the bridge, the bridge columns, and the surfaces on both sides, with further details illustrated in Figure 12d.

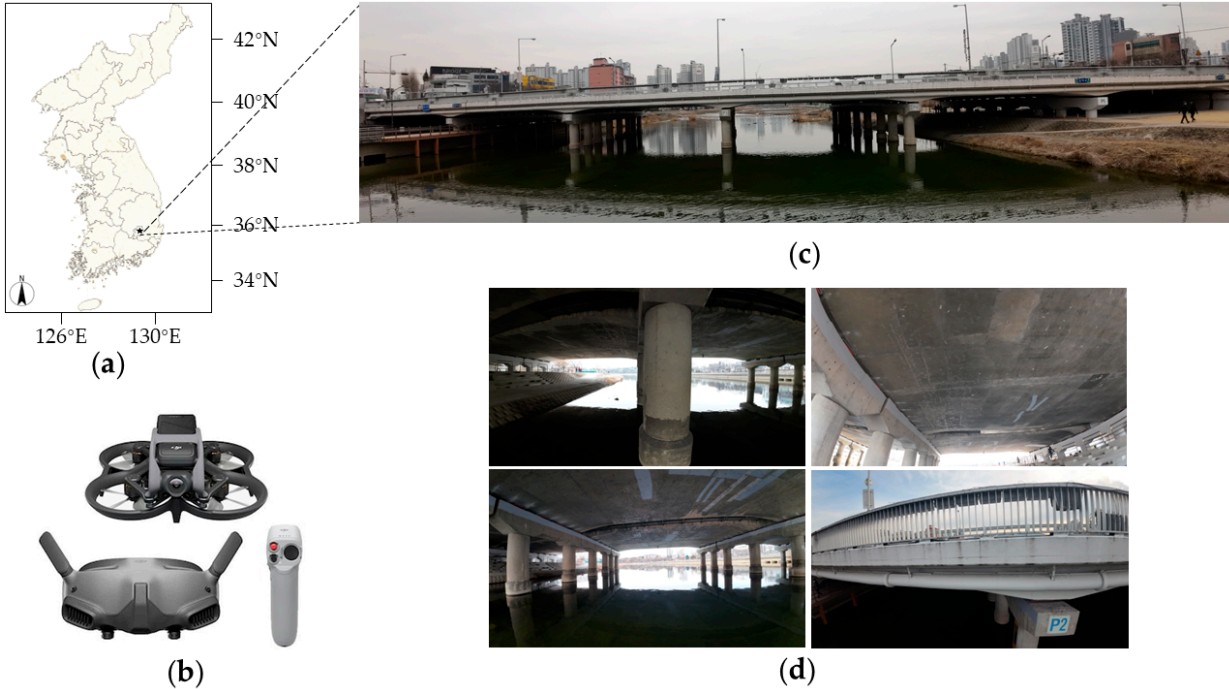

**Figure 12.** The experimental site and inspection path. (**a**) Location of the Kyungdae Bridge test site; (**b**) DJI Avata UAV used for the inspection; (**c**) the inspection path taken by the UAV; (**d**) some samples of specific areas inspected by the UAV.

During the real-time operation, the UAV parameters are shown in Table 2. In consideration of safety and the accuracy of aerial photography, the UAV maintained a distance of 1 m to 1.5 m from the concrete surface for shooting, and the UAV flight attitude is shown in Figure 13. All images were recorded in numerical order. The UAV also automatically recorded the 3D coordinates of the hovering position during the shooting process, which are embedded in the image and can be used to obtain the location of the concrete damage image.

**Table 2.** UAV and experimental parameters.

| UAV Parameters | | Experimental Parameters | |
|---|---|---|---|
| Total Mass | 0.4 kg | Distance maintained (H) | 1 m~1.5 m |
| Size (L×W×H) | $180 \times 180 \times 80$ mm | Pitch angle (R) | $-75~95°$ |
| Maximum Resolution | 4 K/60 fps | Overall time of a single inspection | 18 Minutes |
| Field of View (FOV) | 155° | Number of images | 117 |
| Propeller Protection | Built-in | Wind velocity | 0~3 m/s |

Choosing DJI Avata offers several advantages compared to other UAVs. The compact size of the DJI Avata, combined with its advanced obstacle avoidance and positioning system, makes it highly suitable for data collection in complex environments such as bridges. Additionally, its high-resolution camera and electronic anti-shake technology ensure that it captures high-quality images that are crucial for detecting any damage on the underside of bridges. The DJI Avata drone is equipped with a 3-axis stabilized gimbal and a 20 MP camera capable of capturing high-resolution images and 4K video at 60 frames per second. The camera features a 1-inch CMOS sensor and an adjustable aperture from f/2.8 to f/11, providing greater flexibility in various lighting conditions. Another significant

advantage of the DJI Avata is its gimbal lens design. The gimbal lens is located on top of the body and can be controlled to move at a pitched angle (R), making it easier to capture images of the bottom of bridges than other UAVs. This feature is critical because it not only makes the process more efficient but also allows the UAV to perform inspection tasks more safely and effectively.

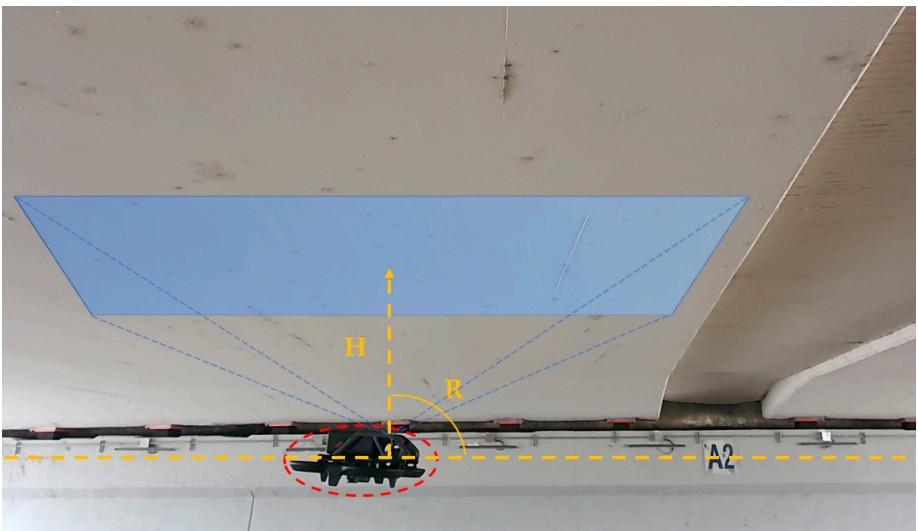

**Figure 13.** Example of the attitude of the UAV while hovering for inspection.

### 2.3. Experiments

### 2.3.1. Experimental Environment and Dataset

The training computer used for the network model had a hardware configuration consisting of an Intel (R) core (TM) i5-11400F CPU, 16 GB of RAM, a Samsung SSD 500 GB, and an NVIDIA GeForce RTX 3050 graphics card. The software configuration included the Windows 11 Pro 64-bit operating system, Python 3.7 programming language, and the Pytorch 1.9.0 learning framework. This hardware and software configuration provided a powerful and efficient computing environment to train and optimize the performance of the network model for object detection tasks.

The COncrete DEfect BRidge IMage Dataset (CODEBRIM) is a collection of high-resolution images of concrete defects on 30 unique bridges [31]. The dataset was created in response to the need for a more diverse set of defect classes, including cracks, spallation, exposed reinforcement bars, efflorescence (calcium leaching), and corrosion (stains) in five categories, which are illustrated in Figure 14. The bridges were selected based on varying overall deterioration, defect extent, severity, and surface appearance. Images were taken under changing weather conditions to include wet or stained surfaces with multiple cameras at varying scales.

The dataset contains 1590 high-resolution images, each with at least one concrete defect, a total of 5354 annotated defect bounding boxes, and 2506 generated non-overlapping background bounding boxes. The training and validation sets were randomly divided into 1272 and 318 images in a ratio of 8:2, and the number of defects for each class is shown in Figure 15. One of the challenges for the proposed approach is that many images contain multiple classes of defects, making the dataset more challenging than a single-class identification task. Additionally, there is significant variation in the aspect ratio, scale, and resolution of the different defects and their bounding boxes, which can vary widely within a single image. By employing these methods, the original dataset of 1272 images for the training set can be effectively expanded to include approximately 12,720 images.

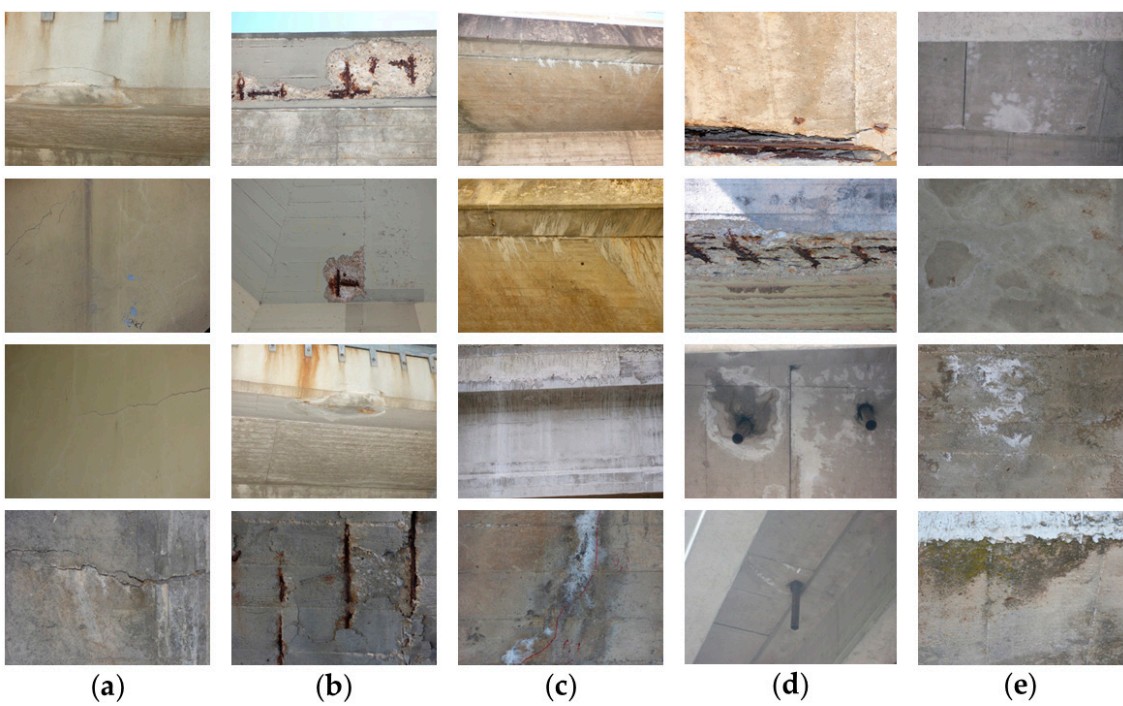

**Figure 14.** Five categories of defects in datasets: (**a**) crack, (**b**) spallation, (**c**) efflorescence, (**d**) exposed bars, and (**e**) corrosion.

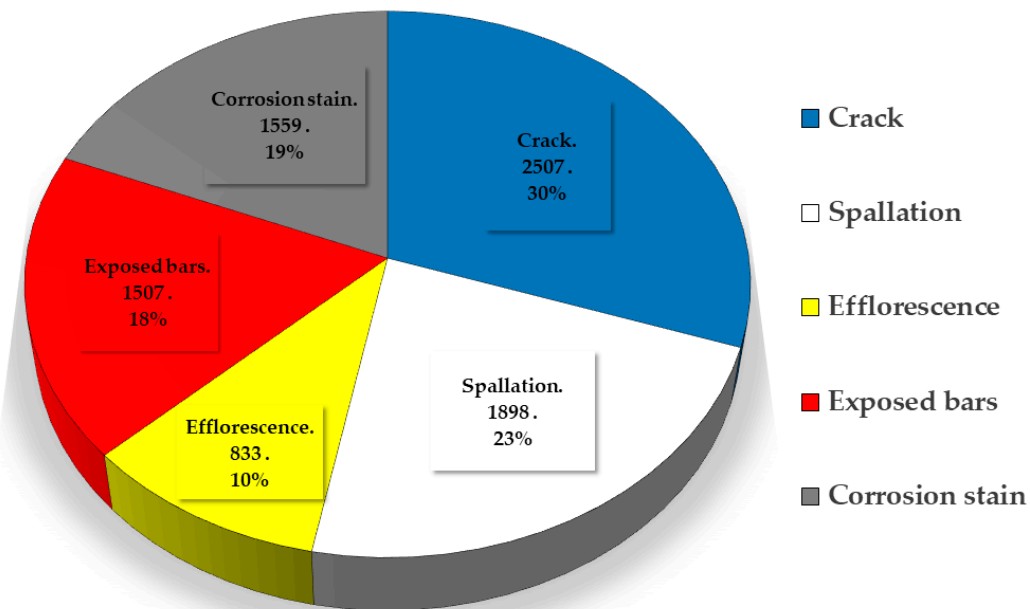

**Figure 15.** The number and percentage of defects in each category.

### 2.3.2. Evaluation Indices and Training Strategies

To assess the effectiveness of the network, several evaluation metrics were introduced, including precision, recall, F1-score, average precision (AP), and mAP. Precision measures the proportion of correctly detected positive cases among all predicted positive cases, as shown in Equation (5), while recall measures the proportion of correctly detected positive cases among all actual positive cases, as shown in Equation (6). F1-score is the harmonic mean of precision and recall, which balances the importance of precision and recall, as shown in Equation (7). AP is the area under the precision-recall curve, which indicates how well the network performs across different levels of precision and recall, as shown in Equation (8). mAP is the average AP calculated over all object classes, which provides

a comprehensive evaluation of the network's performance, as shown in Equation (9). These evaluation indexes, represented mathematically, provide a quantitative measure of the performance of target detection networks and are critical for improving their accuracy and efficiency.

$$\text{Precision} = \frac{\text{TP}}{\text{TP} + \text{FP}}, \tag{5}$$

$$\text{Recall} = \frac{\text{TP}}{\text{TP} + \text{FN}}, \tag{6}$$

$$\text{F1} - \text{score} = 2 \times \frac{\text{Precision} \times \text{Recall}}{\text{Precision} + \text{Recall}}, \tag{7}$$

where true positives (TP) are the number of correctly detected targets, false positives (FP) are the number of non-targets incorrectly identified as targets, and false negatives (FN) are the number of targets that were not detected by the model.

$$\text{AP} = \frac{1}{n} \sum\nolimits_{(r \in \frac{1}{n}, \frac{2}{n} \dots \frac{n-1}{n}, 1)} P_{\text{interop}}(r), \tag{8}$$

$$\text{mAP} = \frac{1}{n} \sum \text{AP}, \tag{9}$$

where $n$ represents the total number of recall levels, which are defined as evenly spaced thresholds between 0 and 1. $r$ represents each recall level, and $P_{interop(r)}$ represents the maximum precision value for any threshold higher than $r$.

We present the hyperparameters used in the training process, as shown in Table 3. The input image size is set to $640 \times 640$, and the batch size is set to 8 to balance training speed and memory usage. The network is trained for a total of 300 epochs, which is sufficient to achieve convergence. We set the maximum learning rate to 0.01 and the minimum learning rate to 0.0001. To optimize the learning rate, we employ the cosine annealing method [32], which reduces the learning rate in a cyclical manner to allow the network to better explore the parameter space. This method has been shown to improve the convergence of deep neural networks and reduce the risk of overfitting. These hyperparameters were selected based on previous research and extensive experimentation to ensure the optimal performance of the network in detecting targets in complex scenes.

**Table 3.** The hyperparameters and the data augmentation method in the training process.

| Input Settings | | | Loss Calculation | | | | Data Enhancement | |
|---|---|---|---|---|---|---|---|---|
| Input shape | Batch size | Total Epoch | Loss Function | Max_lr | Min_lr | Decay Type | Mosaic | Mixup |
| $640 \times 640$ | 8 | 300 | Focal Loss | 0.01 | 0.0001 | Cosine Annealing | True | True |

The two data augmentation methods used in the training-time strategy, mosaic and mixup [33]. The mosaic technique is a data augmentation method that combines multiple images into a single image to train the network. Specifically, four images are randomly selected from the dataset and combined into one larger image. The mixup technique, on the other hand, generates a new training sample by blending two images from the dataset. The method calculates a weighted sum of the pixel values of two images and their corresponding labels to create a new training sample. We utilized both techniques to further enhance the performance of our target detection network.

The loss function used to train the target detection network in this paper was focal loss [34], which is designed to solve the damage class imbalance problem in the detection task. The class imbalance problem occurs when the majority of the training samples belong to the negative class, and the model may have difficulty learning to distinguish the minority positive class. Focal loss reduces the contribution of well-classified examples, which reduces the impact of easy negative samples and focuses on hard examples, thus

effectively addressing the class imbalance problem. The focal loss function is defined as Equation (10):

$$Focal\ Loss(p_t) = -(1 - p_t)^{\gamma} \log(p_t),\tag{10}$$

where $p_t$ is the predicted probability of the correct class for a given input, and $\gamma$ is a focusing parameter that reduces the contribution of easy examples and focuses on hard examples.

## 3. Results

### 3.1. Comparison of the Performance of Each Backbone Network

Firstly, ablation experiments were designed to verify the feasibility and efficiency of the proposed network's backbone. All the essential training requirements, including datasets, hyperparameters, training techniques, and experimental settings, remained constant except for the module parameters. To conduct a comparative analysis, we selected three lightweight networks, namely Shufflenetv2 [35], Mobilenetv2 [36], and Xception [37], along with three larger parametric networks, Densenet121 [38], ResNet50 [39], and VGG16 [40]. It is noteworthy that Shufflenetv2 uses a channel shuffle operation to reduce computation, while Mobilenetv2 focuses on depthwise separable convolutions [41] to achieve a balance between accuracy and efficiency. Xception primarily employs depthwise convolutions and pointwise convolutions to build a deep neural network with high accuracy. On the other hand, Densenet121 emphasizes channel connections to improve the efficiency of feature reuse, while ResNet50 contains a deeper network for higher accuracy. VGG16 is widely used for classical feature extraction with CNN.

Figure 16a shows the convergence state of the loss function for each backbone network after 300 epochs of training for performance comparison. Similarly, Figure 16b shows the mAP curves for the validation set with conservative parameter settings to illustrate the changes in mAP during the training process. It is evident from these figures that our proposed method exhibits superior convergence efficiency and displays the most promising growth trend in terms of mAP.

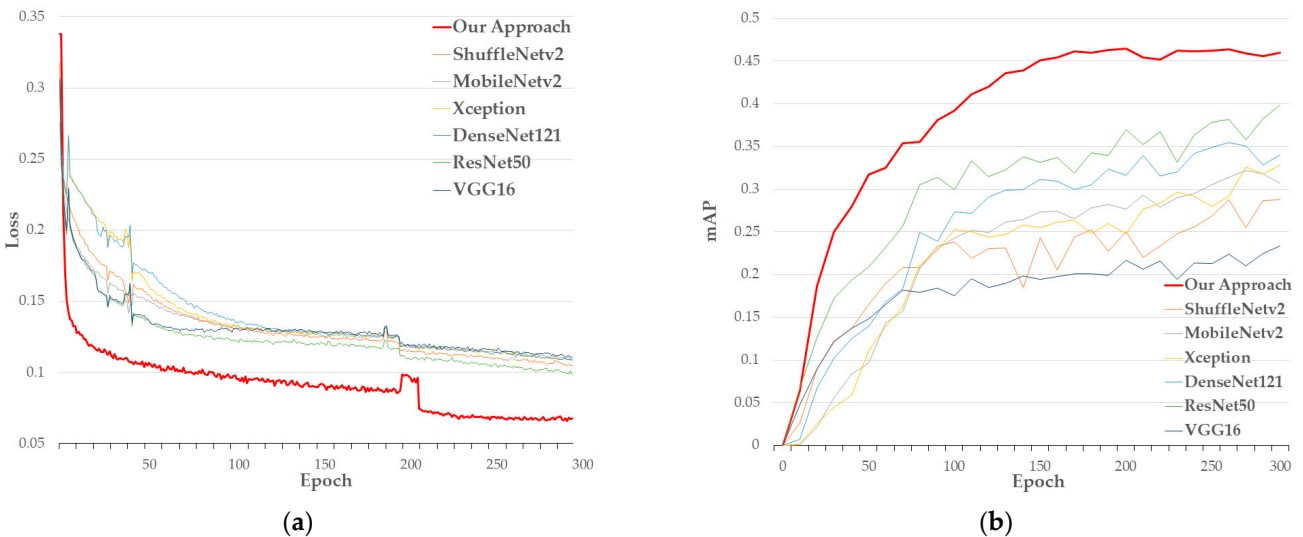

**Figure 16.** Comparison of backbone networks: (**a**) convergence state of loss function, and (**b**) mAP curves.

### 3.2. Comparison of the Attention Modules

Secondly, we comprehensively compared network performance based on different attention mechanisms. As presented in Table 4, where bold numbers indicate the highest mAP, "√" in each column indicates that the leftmost component is used in the model. The last number in each column represents the mAP obtained using the corresponding component. The baseline performance of the network was measured without adding any attention module. We aimed to investigate the efficacy of our proposed attention mechanism, LRGA-Net. Therefore, we selected several popular and representative attention

mechanisms for comparison, including SE-Net [42], ECA-Net [43], CBAM [27], CANet [44], and the LRCA-Net [29]. SENet adapts the feature maps channel-wise to enhance the network's discriminative capability, while ECA-Net integrates global contextual information of the feature maps through channel-wise attention. CBAM captures spatial and channel attention simultaneously, whereas CANet applies attention at multiple scales. In contrast, our proposed LRGA-Net aims to improve the network's local feature learning capability by encoding long-range channel-wise dependencies in the feature maps. We configured these attention mechanisms in the network and compared their performance against the baseline. Figure 17 shows the visual heat map of each output feature layer before and after configuring LRGA-Net in the network. It is evident that after the integration of LRGA-Net, the target is more prominently highlighted in the heat maps, indicating the enhanced effectiveness of LRGA-Net in capturing and emphasizing the target region.

**Table 4.** Network performance of different attention mechanisms.

| | | | | | | | |
|---|---|---|---|---|---|---|---|
| Baseline | √ | √ | √ | √ | √ | √ | √ |
| SENet | | √ | | | | | |
| ECA-Net | | | √ | | | | |
| CBAM | | | | √ | | | |
| CANet | | | | | √ | | |
| LRCA-Net | | | | | | √ | |
| LRGA-Net | | | | | | | √ |
| Parameters (Millions) | 86.01 | 86.49 | 86.15 | 87.78 | 87.52 | 87.41 | 87.3 |
| mAP(%) | 57.49 | 58.57 | 58.99 | 59.74 | 59.63 | 60.77 | 61.27 |

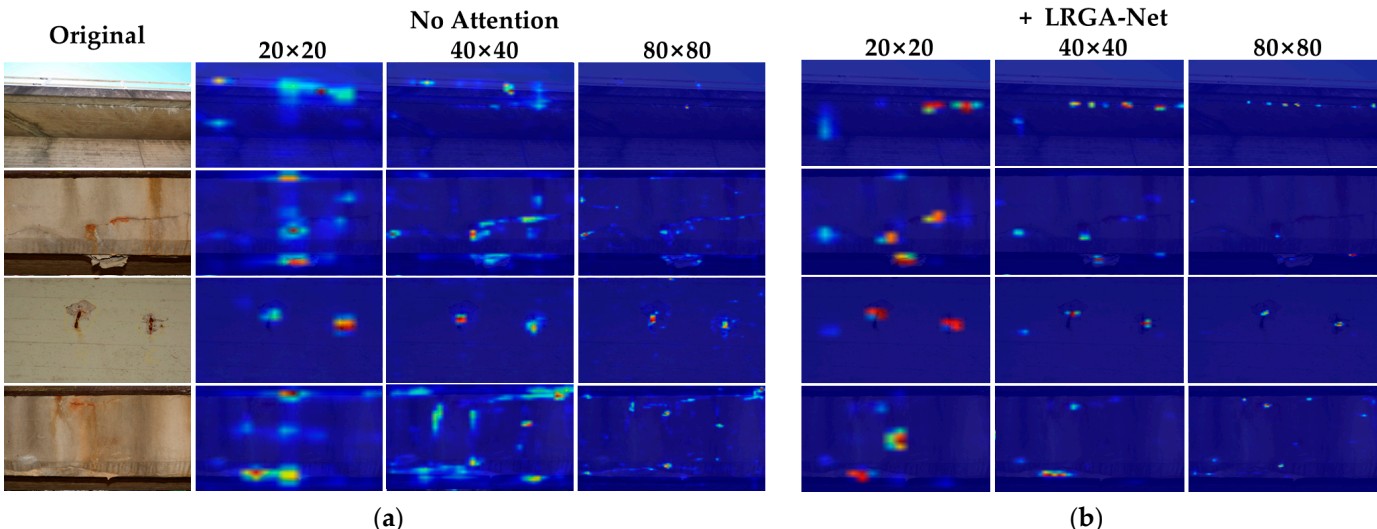

(**a**)           (**b**)

**Figure 17.** Three output sizes of the visual heat map where more highlighted areas indicate higher attention weights given by the network. (**a**) No attention module and (**b**) LRGA-Net.

### 3.3. Comparison of Performance with Other Classical Networks

Finally, we conducted a comparison of our proposed method with some of the widely used object detection networks, including SSD [45], Faster-RCNN [46], YOLOv5, YOLOX [47], and EfficientDet [48]. These models were chosen for comparison because of their popularity and ability to achieve state-of-the-art results in object detection. SSD is a single-shot detector network, whereas Faster-RCNN is a two-stage detector network that includes a region proposal network. YOLOv5 and YOLOX are one-stage detector networks that have been shown to achieve high detection accuracy. EfficientDet is a recent one-stage detector that combines different techniques such as a bi-directional feature pyramid network, compound scaling method, and weighted feature fusion.

We evaluated the performance of all models on the same dataset using the same training method, and the results are shown in Table 5. In addition, we analyzed the accuracy × recall curves for each model and the AP results for each category detection shown in Figure 18, as well as the performance comparison between models shown in Figure 19. Our proposed method exhibited a smoother and broader curve for all detected categories. This characteristic indicates that it attained the highest accuracy across targets of varying sizes. Notably, the proposed method occupies a position in the uppermost region near the middle of the Figure 19. This placement suggests that it strikes a favorable balance between complexity and accuracy, being comparatively less complex while still achieving the highest level of accuracy.

**Table 5.** Comparison of the proposed method and other models on the same dataset, with an interaction of union (IoU) threshold of 0.5.

| Method | | Input Size | Categories-AP | | | | | mAP(%) | F1(%) | Parameters (Millions) | G-FLOPs(G) |
|---|---|---|---|---|---|---|---|---|---|---|---|
| | | | Exposed Bars | Corrosion Stain | Spallation | Crack | Efflorescence | | | | |
| SSD | | 600 × 600 | 0.49 | 0.45 | 0.43 | 0.32 | 0.17 | 37.15 | 5.8 | 26.3 | 247.51 |
| Faster-RCNN | ResNet | 600 × 600 | 0.58 | 0.57 | 0.54 | 0.51 | 0.33 | 50.53 | 17.8 | 135.8 | 374.21 |
| | VGG | 600 × 600 | 0.50 | 0.51 | 0.43 | 0.55 | 0.31 | 46.07 | 15.7 | 29.5 | 932.35 |
| YOLOv5 | L | 640 × 640 | 0.58 | 0.52 | 0.52 | 0.50 | 0.31 | 48.62 | 19.4 | 44.7 | 115.47 |
| | X | 640 × 640 | 0.62 | 0.58 | 0.55 | 0.53 | 0.35 | 52.49 | 21.6 | 83.3 | 218.36 |
| YOLOX | L | 640 × 640 | 0.63 | 0.57 | 0.54 | 0.55 | 0.40 | 53.86 | 23.5 | 35.6 | 109.32 |
| | X | 640 × 640 | 0.64 | 0.60 | 0.58 | 0.56 | 0.41 | 55.91 | 26.4 | 71.8 | 191.47 |
| EfficientDet | D4 | 1024 × 1024 | 0.51 | 0.53 | 0.51 | 0.50 | 0.29 | 46.88 | 13.8 | 20.7 | 113.16 |
| | D5 | 1280 × 1280 | 0.52 | 0.55 | 0.52 | 0.51 | 0.26 | 47.21 | 14.1 | 33.6 | 271.73 |
| | D6 | 1280 × 1280 | 0.58 | 0.57 | 0.53 | 0.53 | 0.27 | 49.67 | 14.8 | 51.8 | 546.46 |
| | D7 | 1536 × 1536 | 0.58 | 0.58 | 0.52 | 0.51 | 0.30 | 49.58 | 14.6 | 57.6 | 655.23 |
| Our Approach | | 640 × 640 | 0.67 | 0.66 | 0.64 | 0.60 | 0.49 | 61.27 | 37.8 | 87.3 | 253.14 |

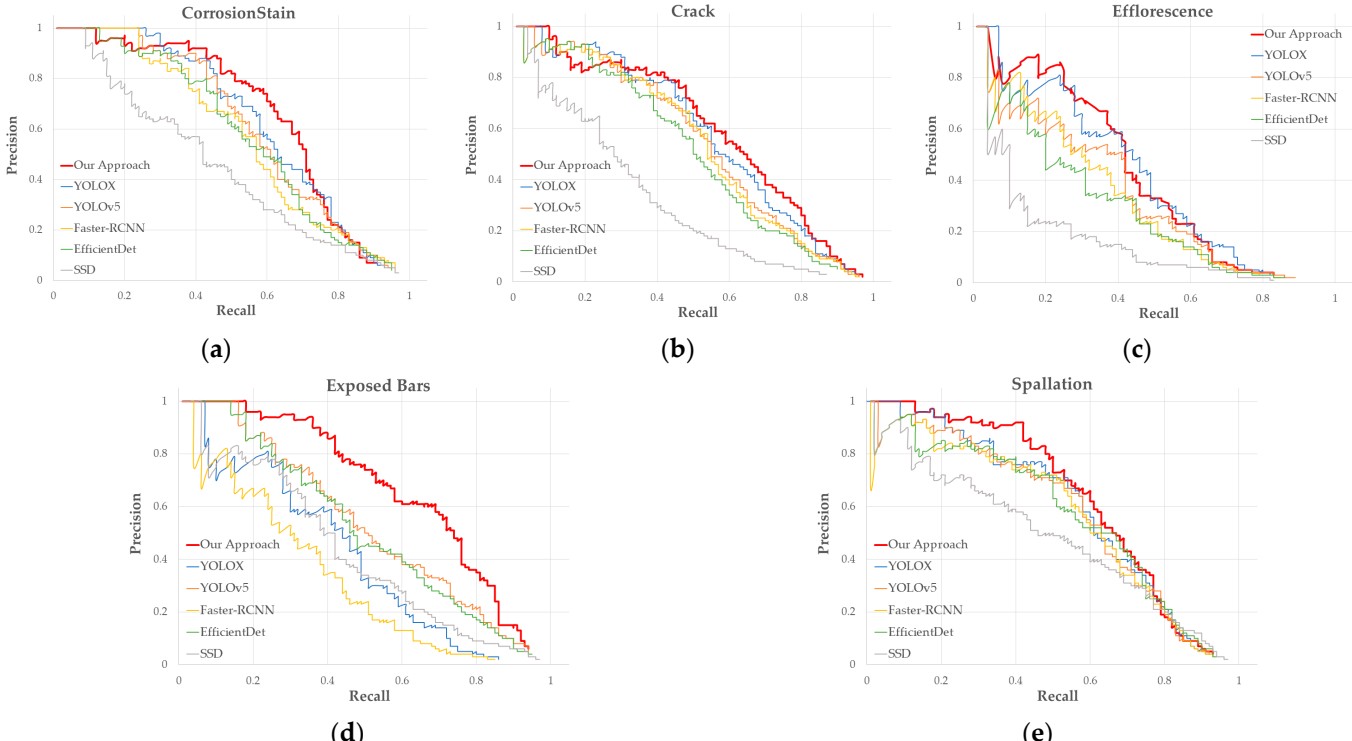

**Figure 18.** Precision × recall curves for each damage category: (**a**) corrosion stain, (**b**) crack, (**c**) efflorescence, (**d**) exposed bars, (**e**) spallation.

In the Table 5, we can observe the performance metrics of our approach compared to other methods. Our approach demonstrated superior performance across multiple categories in terms of AP, mAP, and F1-score.

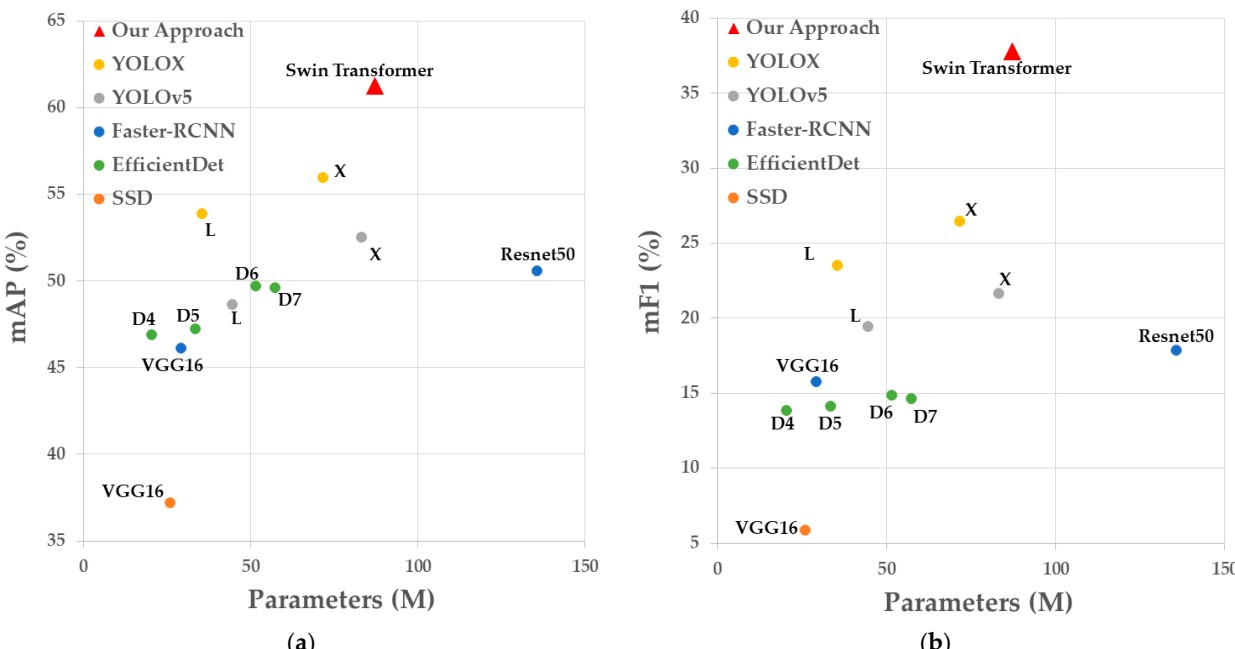

**Figure 19.** Comparison of comprehensive performance of models: (**a**) parameters (M) vs. mAP, (**b**) parameters (M) vs. mF1.

Compared to SSD, Faster-RCNN, and YOLOv5, our approach achieved higher mAP scores, indicating better overall performance. Additionally, our approach outperformed EfficientDet in terms of mAP, F1-score, and parameter efficiency.

Furthermore, our method exhibited similar or even higher AP and mAP scores compared to other YOLOX variants, showcasing its effectiveness in accurately detecting and categorizing exposed bars, corrosion stains, spallation, cracks, and efflorescence.

Notably, our approach stood out with a considerably higher mAP of 0.61, indicating its ability to achieve higher accuracy across all categories. It also showed competitive parameter efficiency, with a relatively lower number of parameters and G-FLOPs compared to some of the other methods listed.

To provide a fair comparison of computational efficiency, we kept the image size of each dataset consistent for both the training and inference phases. We used two metrics, giga floating-point operations per second (G-FLOPs) and frames per second (FPS), to evaluate the computational efficiency of the models. The results of the computational efficiency comparison are illustrated in Figure 20.

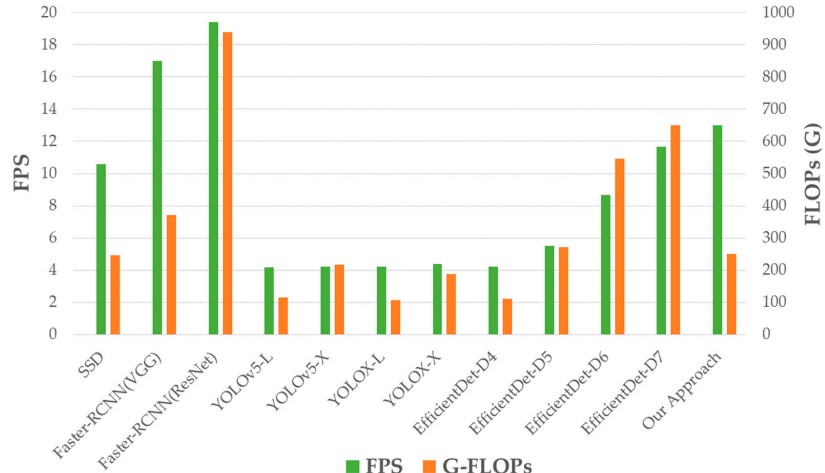

**Figure 20.** Comparison of the computational efficiency of different models.

The findings of this study are presented through Figure 21, which displays a comparison of the visualizations obtained from testing the proposed method and other models using the UAV bridge damage inspection scheme. The obtained image resolution was 3840 × 2160, the maximum suppression method was employed, and the IOU threshold was set to 0.5 to eliminate redundant bounding boxes. This resulted in a video processing speed of approximately 12 frames per second and allowed for a more accurate determination of the type of defect. In Figure 22, we identify instances of false detection where efflorescence was incorrectly identified as spallation. Additionally, there were some cases of missed detection where certain instances were not detected.

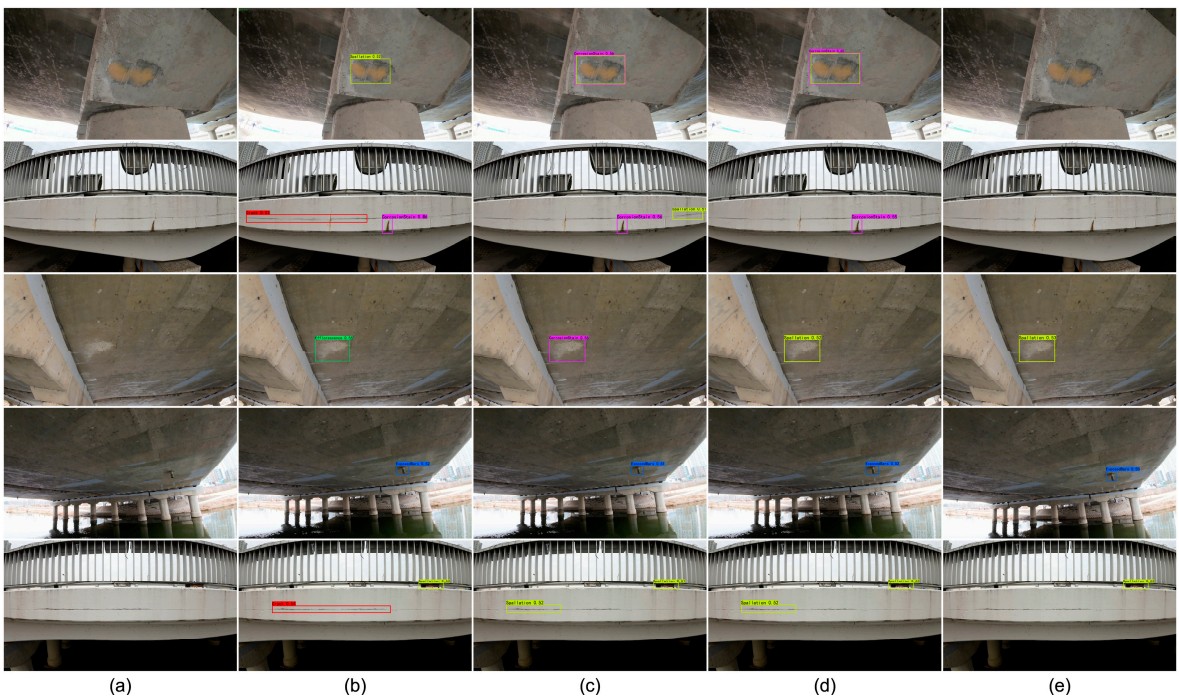

(a)      (b)      (c)      (d)      (e)

**Figure 21.** Sample results of the actual field output of different models. (**a**) Original image, (**b**) our approach, (**c**) YOLOX, (**d**) Faster-RCNN, (**e**) SSD.

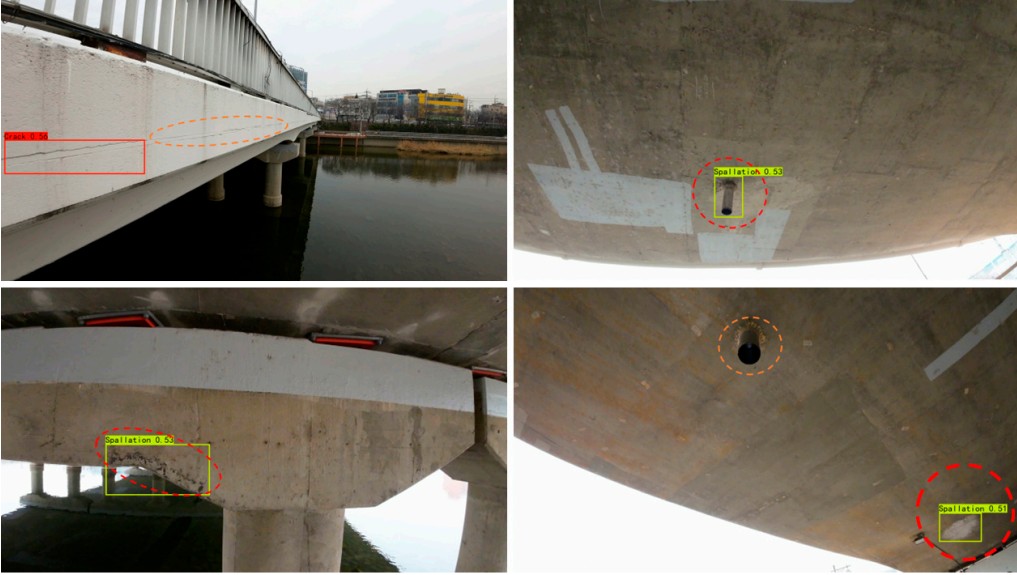

**Figure 22.** Samples of error detection (indicated by red dashed lines) and omission detection (indicated by orange dashed lines) using our proposed model.

## 4. Discussion

The UAV remote sensing-based concrete bridge damage detection system proposed in this paper is inspired by the difficulties of achieving automation in practical inspections. First, the environment constrains the acquisition of bridge damage images, and the manual acquisition process requires a lot of effort and resources. It may not capture hard-to-reach or hidden areas, posing a great risk to the inspection process. In addition, most bridge damage detection efforts are based on pixel-level segmentation methods, which do not allow for the classification and localization of damage types. The paper proposes a bridge damage detection system based on UAV remote sensing to address these issues.

The paper presents the results of ablation experiments comparing different backbone networks. The proposed Swin Transformer-based backbone network showed the best loss function convergence and speed results. We also introduced a proposed attention mechanism, LRGA-Net, and compared it with other representative attention mechanisms in attention pyramid networks. Our LRGA-Net achieved a 3.78% improvement in mAP compared to the baseline. The all-around attention advantage results in better accuracy due to the combination of channel and spatial attention. The heat map visualization significantly improved the detection of targets of all sizes across feature layers. LRGA-Net outperformed channel attention mechanisms such as SENet and ECA-Net in ablation experiments, improving accuracy by 2.7% and 2.28%, respectively. Compared to CBAM, our method overcame the lack of long-distance dependencies associated with a single convolution in the spatial dimension, achieving 1.53% higher accuracy while reducing the computational cost. Compared to the previous LRCA-Net, our method improved accuracy by 0.5%.

We also analyzed the accuracy $\times$ recall curves for each detection category, and our method produced smoother and larger area curves for all classes, indicating the highest accuracy achieved for all types of targets. Our approach was more efficient than other widely used target detection models, such as YOLOv5-L and YOLOX-L, improving mAP by 8.78% and 5.36%, respectively. Although our model's complexity is slightly higher, this investment is worthwhile considering the substantial performance improvement.

Our method achieved higher FPS than other models, such as YOLOv5 and YOLOX, under similar G-FLOPs. We attribute this to two main reasons: on the one hand, G-FLOPs only consider the computation cost but ignore the memory access cost (MAC), which is the time required for memory access. For example, grouped convolution reduces the performance of some optimized implementation algorithms at the lower level due to grouping, resulting in higher time costs as the number of groups increases. In contrast, our Swin Transformer encoder backbone has a linear relationship between its computational complexity and input image size, rather than a quadratic or cubic relationship, which enables it to handle larger image sizes without significantly increasing computation cost and effectively reduce memory access times and data transfer volume. On the other hand, parallelism also affects the speed of the model. Under the same G-FLOPs, models with high parallelism are faster than models with low parallelism. Our method can fully utilize GPU parallel computing capability due to our backbone self-attention mechanism and local window computation characteristics.

Regarding the visual inspection results on the actual construction site, our model exhibited high robustness compared to other methods. Using such techniques and technologies can significantly enhance the accuracy and speed of bridge damage detection, thereby improving the overall maintenance and safety of bridge infrastructure. However, we observed false detection in Figure 22, incorrectly detecting efflorescence as spallation and some missed detection cases. This may be caused by the inconsistency between the image target size and the shooting angle in the training dataset; the difference between the presented efflorescence and corrosion stain in the dataset is not apparent.

In summary, the proposed bridge damage inspection system based on UAV remote sensing addresses many of the limitations of traditional inspection methods. The excellent performance and accuracy achieved in this study demonstrate the potential of this method to revolutionize bridge inspection and maintenance. However, more work is needed to

address the current model's limitations, particularly in improving the training dataset to enhance the model's robustness and ability to detect damage accurately.

## 5. Conclusions

This paper proposes a UAV remote sensing-based concrete bridge damage detection system that overcomes the limitations of traditional inspection methods. The results of the ablation experiments demonstrate the superiority of the proposed Swin Transformer-based backbone network, showcasing better loss function convergence and speed compared to other networks. The introduced attention mechanism, LRGA-Net, achieves significant improvements in mAP by combining channel and spatial attention, and surpassing widely used detection models like YOLOv5-L and YOLOX-L. Despite a slightly higher complexity, the investment in the proposed method is justified by the substantial performance improvement. The visual inspection results on actual construction sites demonstrate the high robustness of the proposed model compared to other methods. By enhancing the accuracy and speed of bridge damage detection, the proposed system has the potential to significantly improve the overall maintenance and safety of bridge infrastructure. However, some false detections and missed cases were observed, potentially attributed to inconsistencies between target size and shooting angles in the training dataset, as well as subtle differences between presented efflorescence and corrosion stains. Considering the limitations, we believe that a strategy for overcoming these drawbacks in subsequent studies involves updating the dataset as needed. This entails improving the quality and relevance of the dataset, ensuring that it better aligns with the specific challenges encountered in the practical deployment of the proposed system. Simultaneously, a rational development of network models should be pursued, aiming to enhance their adaptability to the diverse scenarios encountered in real-world bridge inspections. Overall, this system has the potential to revolutionize bridge inspection and maintenance and improve overall infrastructure safety.

**Author Contributions:** Conceptualization, H.L. and S.-C.L.; methodology, H.L.; software, H.L. and S.S.; validation, H.L.; formal analysis, H.L.; investigation, H.L., S.-C.L. and S.S.; resources, H.L. and S.S.; data curation, H.L.; writing—original draft preparation, H.L.; writing—review and editing, S.-C.L. and S.S.; visualization, H.L.; supervision, S.-C.L. and S.S.; project administration, S.-C.L. and S.S.; funding acquisition, S.-C.L. All authors have read and agreed to the published version of the manuscript.

**Funding:** This research was funded by Basic Science Research Program through the National Research Foundation of Korea (NRF) funded by the Ministry of Education (NRF-2020R1I1A3073831).

**Data Availability Statement:** Not applicable.

**Conflicts of Interest:** The authors declare no conflict of interest.

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
