# Peer review of "UAV-Based Low Altitude Remote Sensing for Concrete Bridge Multi-Category Damage Automatic Detection System"

_drones, doi:10.3390/drones7060386_

Round 1

Reviewer 1 Report

An interesting paper. The authors did excellent work. I added a few comments and suggestions that could help to improve the paper. 

- Check lines 294 to 298. Hard to follow.

- Figures 16-22 require a better explanation. Include a complete analysis in their corresponding sections.

- Need to include a figure showing the confusion matrice. I suggest including the figure.

- Table 5 requires a thorough analysis; include the study.

- The conclusion needs a summary of the results obtained (%). 

- The paper mentioned the future work needed; please explain what has been planned to be done in the future.

Author Response

Responses are in the document, thank you.

Reviewer 2 Report

Could you provide further details on the shortcomings of the suggested system, including false positive and false negative detections arising from discrepancies within the training data? What are your strategies for overcoming these drawbacks in subsequent studies?

How do you envision this system being implemented in practice? What are some potential barriers to adoption and how can they be addressed? Please add explanations.

How do you ensure that the proposed system is not biased against certain types of bridges or damage categories? Have you considered any measures to mitigate potential biases? Please add explanations.

It is needed to add more reference regarding the detection/recording bridge damage by 3d model. including:
https://scholar.google.co.jp/scholar?hl=ja&as_sdt=0%2C5&q=detecting+3d+bridge+damage&btnG=

Author Response

(The authors gave the same response as above.)

Reviewer 3 Report

This manuscript proposes a novel UAV-based concrete bridge damage detection system that utilizes a Swin Transformer-based backbone network and LRGA-Net. Overall, the manuscript is well-written, and the research is well-designed. It could be accepted after minor revisions are made.

In line 104, the abbreviation "(mAP)" should be added after "mean Average Precision" when it is mentioned for the first time.

In lines 360-361, the training and validation sets are divided in an 8:2 ratio, and then in lines 398-405, it is mentioned that data augmentation is applied. The manuscript should specify the final number of images actually used after these processes. This detail would provide more clarity regarding the dataset size and composition.

The conclusions section requires improvement for enhanced clarity and impact.

Author Response

(The authors gave the same response as above.)
